# Exploiting Potential Probiotic Lactic Acid Bacteria Isolated from *Chlorella vulgaris* Photobioreactors as Promising Vitamin B12 Producers

**DOI:** 10.3390/foods12173277

**Published:** 2023-09-01

**Authors:** Mónica Ribeiro, Cláudia Maciel, Pedro Cruz, Helena Darmancier, Teresa Nogueira, Margarida Costa, Joana Laranjeira, Rui M. S. C. Morais, Paula Teixeira

**Affiliations:** 1CBQF—Centro de Biotecnologia e Química Fina—Laboratório Associado, Escola Superior de Biotecnologia, Universidade Católica Portuguesa, Rua Diogo Botelho 1327, 4169-005 Porto, Portugalrcmorais@ucp.pt (R.M.S.C.M.); 2INIAV—Instituto Nacional de Investigação Agrária e Veterinária, I.P., 2780-159 Oeiras, Portugal; 3cE3c—Center for Ecology, Evolution and Environmental Changes & CHANGE—Global Change and Sustainability Institute, 1749-016 Lisbon, Portugal; 4ALLMICROALGAE Natural Products S.A., R&D Department, Rua 25 de Abril s/n, 2445-413 Pataias, Portugal

**Keywords:** cobalamin, B12 biosynthesis, B12-producing probiotic lactic acid bacteria, *C. vulgaris*

## Abstract

Lactic acid bacteria (LAB) have been documented as potential vitamin B12 producers and may constitute an exogenous source of cobalamin for the microalga *Chlorella vulgaris*, which has been described as being able to perform vitamin uptake. Hence, there is an interest in discovering novel B12-producing probiotic LAB. Therefore, the purpose of the current work was to perform a phenotype–genotype analysis of the vitamin B12 biosynthesis capacity of LAB isolated from *C. vulgaris* bioreactors, and investigate their probiotic potential. Among the selected strains, *Lactococcus lactis* E32, *Levilactobacillus brevis* G31, and *Pediococcus pentosaceus* L51 demonstrated vitamin B12 biosynthesis capacity, with the latter producing the highest (28.19 ± 2.27 pg mL^−1^). The genomic analysis confirmed the presence of pivotal genes involved in different steps of the biosynthetic pathway (*hemL*, *cbiT*, *cobC*, and *cobD*). Notably, *P. pentosaceus* L51 was the only strain harboring *cobA*, *pduU*, and *pduV* genes, which may provide evidence for the presence of the cobalamin operon. All strains demonstrated the capability to withstand harsh gastrointestinal conditions, although *P. pentosaceus* L51 was more resilient. The potential for de novo cobalamin biosynthesis and remarkable probiotic features highlighted that *P. pentosaceus* L51 may be considered the most promising candidate strain for developing high-content vitamin B12 formulations.

## 1. Introduction

Humans acquire vitamin B12 from dietary sources, mainly through consuming foods of animal origin, particularly ruminants and fish, since the vitamin biosynthesized by bacteria accumulates in animal tissues [1]. Suboptimal serum levels of this vitamin may result in B12 hypovitaminosis, a global condition that has raised an increasing interest in developing dietary/pharmacological solutions, such as biofortified foods, with enhanced content in cobalamin [2]. Vitamin B12 deficiency may arise from health conditions such as pernicious anemia and food-bound cobalamin malabsorption (FBCM) or insufficient intake [3]. It is also commonly found in elderly people and individuals with restrictive diets associated with vegetarianism [4,5,6]. The consequences may be observed at the hematological level, including macrocytic or megaloblastic anemia, leukopenia, pancytopenia, thrombocytopenia, and thrombocytosis, or neurological level, such as peripheral neuropathy and areflexia, and in cases of a more severe deficiency, can lead to dementia and psychosis [3,7].

Vitamin B12 represents a cobalt corrinoid family being found in nature mainly in two forms: adenosylcobalamin and methylcobalamin, also referred to as coenzyme B12. The term cyanocobalamin refers to the industrially produced form. In nature, biosynthesis can occur through a de novo pathway, via an oxygen-dependent or independent mechanism, or through the salvage pathway [8]. The first steps of vitamin synthesis rely on the tetrapyrrole synthesis pathway, in which 5-aminolevulinic acid (ALA), the first committed precursor, is synthesized. Following uroporphyrinogen III synthesis, this molecule is subsequently converted into precorrin-2 [8,9,10]. At this stage, the pathways diverge, and the cobalt ion is added at different phases [8,11]. The oxygen-independent pathway is characterized by the early insertion of the cobalt ion into precorrin-2, which results in a high number of cobalt complexes as intermediates [8]. The oxygen-dependent pathway is characterized by later chelation, which is only verified nine steps later [8,9].

Cobalamin is commercially produced through fermentation processes performed by selected microorganisms, well recognized as efficient producers [8,12,13]. *Propionibacterium* spp. strains are the most appealing bacterial candidates for industry, owing to their intrinsic capability to biosynthesize high levels of vitamin B12, and GRAS (Generally Recognized As Safe) status [8]. There is a significant interest in exploiting lactic acid bacteria (LAB) for cobalamin production due to their ubiquity in the environment (in particular, the predominance of this bacterial community in fermented foods), and probiotic traits. The first LAB found to produce a cobalamin analogue was *Limosilactobacillus reuteri* CRL 1098, and this compound was identified as pseudocobalamin [14,15]. After this discovery, the cobalamin biosynthesis cluster was described for this strain [16] and later, this cluster was also identified in *Furfurilactobacillus rossiae* DSM 15814 genome [17]. Moreover, the Biogaia strain *Lim. reuteri* ATCC 55730 was also found to harbor the *pdu*-*cob* cluster and some cobalamin biosynthesis genes were identified in *Lentilactobacillus buchneri* ATCC 11577, *Lentilactobacillus hilgardii* ATCC 8290, *Loigolactobacillus coryniformis* KCTC 3167, *Lo. coryniformis* KCTC 3535, and *Levilactobacillus brevis* ATCC 27305 [18,19]. The cobalamin biosynthesis capacity was found to be widespread within the genus *Lactobacillus*, being identified in some strains of *Lo. coryniformis*, *Lactiplantibacillus plantarum*, and *Limosilactobacillus fermentum* species [20,21,22,23,24].

Despite the assumption that vitamin B12 biosynthesis is restricted to certain prokaryotic organisms, a study performed by Watanabe et al. [25] provided some evidence that eukaryotic green microalgae, like *Chlorella vulgaris*, might be capable of producing biologically active cobalamin analogues. Moreover, the authors demonstrated that *C. vulgaris* possessed the ability to perform the exogenous uptake of vitamin B12.

In light of this finding and based on the previous detection of vitamin B12 in *C. vulgaris* grown in photobioreactors, an interest in exploiting a potential exogenous microbial source of cobalamin has emerged. Hence, the purpose of the present work was to isolate LAB from those *C. vulgaris* cultures, perform a phenotypic evaluation and genotypic confirmation of their vitamin B12 biosynthesis capacity, and investigate the probiotic potential.

## 2. Materials and Methods

### 2.1. Isolation of Lactic Acid Bacteria from Photobioreactor’s Microbiota

Fresh samples of a *C. vulgaris* culture industrially grown in autotrophic conditions in tubular photobioreactors (PBRs) were supplied by Allmicroalgae Company—Natural Products S.A. (Pataias, Portugal). At least three growing PBRs were analyzed. These samples were 10-fold serially diluted in quarter-strength Ringer’s solution (Biokar Diagnostics, Beauvais, France), and 100 µL of appropriate dilutions was spread onto de Man, Rogosa, and Sharpe (MRS) Agar (Biokar Diagnostics) [26]. After 72 h of incubation at 30 °C, 20% of total colonies were randomly selected from countable agar plates [27] and cultured on new MRS agar plates. The obtained isolates were examined by Gram reaction and catalase activity test. The cell morphology was inspected under an optical microscope.

### 2.2. Genetic Identification of Bacterial Strains

The genetic identification of the selected isolates was performed by 16S rRNA sequencing. Isolated colonies were transferred to MRS broth (Biokar Diagnostics), and DNA extraction was performed using the GRS Genomic DNA Kit Bacteria (GRiSP Research Solutions, Porto, Portugal) following the manufacturer’s instructions. A polymerase chain reaction (PCR) assay for the amplification of 16S rRNA gene was carried out according to Lane [28] and Ferreira Da Silva et al. [29]. The reaction was performed in a T100 thermocycler (Bio-Rad Laboratories, Hercules, CA, USA) under the following conditions: 5 min at 94 °C, followed by 30 cycles of 30 s at 94 °C, 30 s at 55 °C, 1.5 min at 72 °C, and a final extension at 72 °C for 1.5 min.

Sequencing was outsourced to Eurofins Genomics company (Ebersberg, Germany). Sequence analysis were performed using the Geneious Prime version 2020.0.5 software (Biomatters, West Auckland, New Zealand).

### 2.3. Determination of Vitamin B12 Biosynthesis by Lactic Acid Bacteria

To determine cobalamin production, a microbiological assay was performed as formerly described [30], with slight modifications. In brief, the novel isolated cultures and positive and negative control strains (*Fur. rossiae* DSM 15814 and *Lpb. plantarum* DSM 20205, respectively) were inoculated into vitamin B12-free assay medium (Merck, Buenos Aires, Argentina). After incubation for 18 h at 30 °C, 50 µL of the culture was transferred to 5 mL of fresh B12-free medium. This procedure was repeated every 18 h, until eight subcultures were achieved. The last subculture was submitted to 115 °C for 10 min, to disrupt the cell membrane and release the intracellular content, and then filtered through sterile 0.2 µm pore-size filters (Minisart Syringe Filter, Sartorius Stedim Biotech GmbH, Goettingen, Germany) to remove cell debris (the resulting media contained the cobalamin biosynthesized by each isolated strain or control cultures, and were denominated cell extract and supernatant media, CESM). In parallel, a vitamin B12 (Merck) stock solution was used to prepare vitamin B12-free assay medium supplemented with cobalamin concentrations ranging from 0 to 100 pg mL^−1^, in order to obtain a standard curve. All tubes were submitted to the same temperature treatment as the test and control tubes. *Lactobacillus leichmannii* DSM 20355 (formerly *Lactobacillus delbrueckii* subsp. *lactis*), a strain that requires vitamin B12 to grow [31], was cultured in MRS broth for 24 h at 37 °C in microaerophilic conditions and transferred two times in the same media. The bacterial culture was centrifuged at 5000× *g*, for 10 min, and washed three times in Ringer’s solution. The bacterial cell density was adjusted to 10^4^ colony-forming units (CFU) mL^−1^ through absorbance measurement at 600 nm, and the cellular suspension was inoculated (1%, *v*/*v*) into the vitamin B12-free assay medium supplemented with different cobalamin concentrations or CESM (media in which each isolated LAB or control strains previously grew). After 72 h of incubation at 37 °C in microaerophilic conditions, *L. leichmannii* culture grown in each medium was serially diluted and inoculated on MRS agar for 48 h at 37 °C in microaerophilic conditions to determine bacterial cell numbers (CFU mL^−1^). Thereafter, a standard curve correlating cobalamin concentration with cell numbers of *L. leichmannii* was generated and, through interpolation, the vitamin B12 contents of CESM collected for each LAB or control strain were determined.

### 2.4. Next-Generation Sequencing (NGS) and Bioinformatics Analysis

*Lactococcus lactis* E31, *Lev. brevis* G31, *Lpb plantarum* G12, and *P. pentosaceus* L51 genomes were sequenced at Eurofins genomics, by a Genome Sequencer Illumina HiSeq technology, using NovaSeq 6000 S2 PE150 XP sequencing mode. Paired-end fastq files were used to assemble the bacterial genomes at INIAV (Instituto Nacional de Investigação Agrária e Veterinária, I. P.) bioinformatics server, using SPAdes (St. Petersburg genome assembler) version SPAdes-3.12.0-Linux [32].

The derived FASTA files of protein sequences were used to search for orthologous proteins involved in cobalamin synthesis. Protein sequence alignments were performed with Blastp, using ncbi-blast-2.10.1 + -1.x86_64.rpm version (downloaded on 8 June 2020) in a Linux environment [33] against a database composed of a protein FASTA file of the partial sequence of *Lim. reuteri* CRL 1098 cobalamin gene cluster, deposited with accession no. AY780645.1 at NCBI (https://www.ncbi.nlm.nih.gov/nuccore/AY780645.1/, accessed on 25 January 2021), using a cut-off e-value of <10^−4^ and an identity of nearly 30%. Blastx alignments were also performed against the Virulence Factor Database (VFDB) (downloaded from http://www.mgc.ac.cn/VFs/Down/VFDB_setB_pro.fas.gz, accessed on 6 November 2020) with an e-value <10^−4^ and a coverage higher than 60%. Acquired antibiotic resistance genes were searched with the ResFinder software (https://cge.food.dtu.dk/services/ResFinder/, accessed on 6 November 2020) using the default parameters of ≥80% identity over ≥60% of the length of the target gene.

### 2.5. Assessment of Lactic Acid Bacteria Probiotic Potential

#### 2.5.1. In Vitro Gastrointestinal Survival Analysis

Resistance of the selected LAB to the simulated gastrointestinal tract (GIT) conditions was evaluated in the static in vitro gastrointestinal digestion model described by Brodkorb et al. [34]. The digestion procedure was conducted in a shaking (200 rpm) water bath at 37 °C. At predefined time intervals, for enumeration of viable bacterial cells, aliquots were collected, serially 10-fold diluted in sterile phosphate buffer saline (PBS, pH 7.4, Sigma-Aldrich St. Louis, MO, USA) and plated in duplicate, using the drop plate method, onto MRS agar, and incubated at 30 °C for 48 h. Gastrointestinal survival analysis was performed in three independent experiments.

#### 2.5.2. Evaluation of the Safety Profile of B12-Producing Strains

##### Antibiotic Susceptibility Evaluation

Minimal inhibitory concentrations (MIC) of different antibiotics were determined using the broth microdilution method following the guidelines of the Clinical and Laboratory Standards Institute (CLSI) and the European Committee on Antimicrobial Susceptibility Testing (EUCAST). For this purpose, the antimicrobial susceptibility of *Lac. lactis*, *Lev. brevis*, *Lpb. Plantarum*, and *P. pentosaceus* towards ampicillin, vancomycin, gentamycin, kanamycin, streptomycin, erythromycin, clindamycin, tetracycline, and chloramphenicol was evaluated. *Enterococcus faecalis* ATCC 29212 was used as a quality control reference strain. The antibiotics selected and the epidemiological cut-off values considered were those established by the European Food Safety Authority (EFSA) Panel on Additives and Products or Substances used in Animal Feed [35].

##### Virulence Genes, Hemolytic Activity, and Biogenic Amine Production

The selected isolates (*Lac. lactis* E32, *Lev. brevis* G31, *Lpb. plantarum* G12, and *P. pentosaceus* L51) were evaluated for the presence of virulence genes (*agg*, *esp*, *gelE*, *efaAfs*, *efaAfm*, *cylA*, *cylB*, *cylM*, *cylL_L_*, and *cylL_S_*) by PCR. Primer sequences and PCR conditions were based on those previously reported by Eaton and Gasson [36] and Semedo et al. [37]. *Enterococcus faecalis* P1 strain was used as positive control for *agg* and *gelE* genes, *E. faecalis* F2 for *cylA*, *cylB*, *cylM*, and *efaAfs* genes, *E. faecalis* DS16 for *cylL_L_* and *cylL_S_* genes, and *E. faecium* F10 and *E. faecalis* P36 to *efaAfm* and *esp* genes, respectively. Positive-control strains were obtained from the culture collection of Tracy Eaton (Division of Food Safety Sciences, Institute of Food Research, Norwich, UK), with the exception of *E. faecalis* DS 16 (from culture collection of C.B. Clewell, Department of Oral Biology, School of Dentistry, University of Michigan, Ann Arbor, MI, USA).

The hemolytic activity of the selected isolates was determined according to Semedo et al. [37].

Biogenic amine production was evaluated as described by Bover-Cid and Holzapfel [38].

### 2.6. Statistical Analysis

All statistical analyses were performed using IBM SPSS Statistics 27.0 (New York, United States). Analysis of variance (ANOVA) was utilized, with Tukey’s test as post hoc, when data were normally distributed, and homoscedasticity of variances was met. Otherwise, non-parametric tests were performed as substitutes, namely, Kruskal–Wallis and independent t-test, respectively. All analyses were performed with a significance of 0.05.

## 3. Results and Discussion

### 3.1. Identification of Lactic Acid Bacteria

Among the 650 LAB collected from cultures of *C. vulgaris* grown in three PBRs, 130 isolates (Gram-stain positive, catalase-negative) were randomly selected. A total of 95 of those isolates were identified as *Lac. lactis* (short chain cocci-shaped cells), 16 as *E. casseliflavus* (paired coccoid-shaped cells), 14 as *Lpb. plantarum* (bacilli-shaped cells), three as *P. pentosaceus* (spherical-shaped quadruplet cells), and two as *Lev. brevis* (rod-shaped cells) (Table 1).

Four isolates (one representative strain of each LAB species) were randomly selected to evaluate their capacity to biosynthesize vitamin B12, namely, *Lac. lactis* E32, *Lev. brevis* G31, *Lpb. plantarum* G12, and *P. pentosaceus* L51. *Enterococcus casseliflavus* isolates were excluded from this analysis, owing to the raised concerns pertaining to their association with human infections [39,40].

### 3.2. Evaluation of Vitamin B12 Biosynthesis by Lactic Acid Bacteria

The selected LAB isolates were screened for their vitamin B12 biosynthesis capacity following a microbiological approach for quantification. The results indicated that *Lac. lactis* E31, *Lev. brevis* G31, and *P. pentosaceus* L51 are capable of producing vitamin B12: 6.18 ± 1.08 pg mL^−1^, 5.47 ± 0.37 pg mL^−1^ and 28.19 ± 2.27 pg mL^−1^, respectively (Figure 1). Moreover, *P. pentosaceus* L51 achieved a production level equivalent (*p* > 0.05) to *Fur. rossiae* DSM 15814 (29.30 ± 1.21 pg mL^−1^), the strain used as the positive control. *Lactococcus lactis* E31 and *Lev. brevis* G31 were found to biosynthesize cobalamin at the same level. In contrast to these results, *Lpb. plantarum* G12 was considered a non-producer strain since the detected levels of cobalamin (0.21 ± 0.17 pg mL^−1^) were similar (*p* > 0.05) to those determined for the negative control (*Lpb. plantarum* DSM 20205). Residual cobalamin production (0.32 ± 0.05 pg mL^−1^) by *Lpb. plantarum* DSM 20205 could be because the indicator microorganism *L. leichmannii* DSM 20355, used in the microbiological assay for vitamin B12 detection, can replace cobalamin with deoxyribosides and deoxynucleotides (the alkali-resistant factor) or the growing of pseudocobalamin, leading to an overestimation of concentration values [41].

*Levilactobacillus brevis* and *P. pentosaceus* have been documented to produce between 0.4 to 0.6 pg mL^−1^ (Table 2) [21]. Although *Lpb. plantarum* G12 had no biosynthesis capacity, other *Lpb. plantarum* strains have been identified as potential cobalamin producers. In the study conducted by Masuda et al. [21], *L. plantarum* CN-225 produced 2.0 pg mL^−1^ of vitamin B12. A *Lpb. plantarum* strain isolated from kanjika showed a cobalamin production of 13 ng g^−1^ of dry biomass when determined by an ELISA method [42]. *Lactiplantibacillus plantarum* strains BHM10, isolated from human milk, and BCF20, isolated from child feces, showed a production between 0.5 and 0.8 pg mL^−1^, after growth in a vitamin B12 assay medium [24]. To the best of our knowledge, no other study reports a *Lac. lactis* strain capable of producing B12.

Further studies must be conducted to discriminate the cobalamin analogue produced by the selected strains. The production of pseudocobalamin by *Lim. reuteri* CRL1098 has already been reported by Santos et al. [15], and this analogue is not bioavailable for humans and other mammals [43].

### 3.3. Genotypic Confirmation of Vitamin B12 Biosynthesis

The phenotypic results obtained were confirmed by comparative genomic analysis against the *Lim. reuteri* CRL 1098 cobalamin synthesis gene cluster, which has previously been described by Santos et al. [16]. Four genes involved in different steps of the biosynthetic pathway were identified in *Lac. lactis* E32, *Lev. brevis* G31, and *Lpb. plantarum* G12 genomes (Table 3): *hemL*, *cbiT*, *cobD*, and *cobC.* The gene *hemL* is involved in uroporphyrinogen III synthesis; *cbiT* and *cobD* are involved in adenosylcobinamide synthesis; and *cobC* is involved in lower ligand synthesis. Although *hemL* was not found in *P. pentosaceus* L51, *cobA*, another gene involved in adenosylcobinamide synthesis, was also identified.

The use of a cut-off of expected value below 10^−4^, and an identity of nearly 30% (which is adequate for protein alignments) suggest these orthologues’ presence, with the exception of corrinoid adenosyltransferase, encoded by *cobA*, which showed an identity of 38%. Moreover, only a few genes involved in cobalamin biosynthesis were found. Nonetheless, other genes related to vitamin biosynthesis, which are present in the same cluster, were detected. For instance, all strains harbored *cbiO*, which is related to the biosynthetic pathway since this gene codifies a cobalt import ATP-binding protein.

Propanediol dehydratase requires cobalamin as a cofactor, and the association of both *pdu* and *cbi-cob* clusters in the genome can emerge from this requirement [44]. *Pediococcus pentosaceus* L51, found to be the most promising cobalamin producer, was the only strain harboring *pduU* (e-value = 2.00 × 10^−32^; 86% identity) and *pduV* genes (e-value = 4.00 × 10^−48^; 55% identity), which codify propanediol utilization proteins. This can be considered as evidence that the cobalamin operon is present in the genome of this strain.

Taken all together, these results do not allow concluding about the biosynthetic pathway on the subject strains. However, it must be considered that the genome sequencing was performed with a short-read approach that faces some challenges in de novo assembly. Although Illumina generates accurate reads, sequencing errors may still occur, mostly in regions with high GC and AT content, and lead to misassembling. Moreover, the need to resort to PCR results in uneven read depth and, ultimately, in the introduction of gaps during assembly. Finally, the presence of repetitive sequences in genomes can also present a problem since, in general, sequences generated are too short to cover an entire repetitive region. Therefore, all repeated sequences may form a chimeric contig. These limitations could be overcome by using a complementary long-read method, which produces larger contigs and is not PCR-biased. In this scenario, a hybrid approach is used, and short reads are assembled with long contigs [45,46].

Genes involved in cobalamin biosynthesis were detected in *Lpb. plantarum* G12, for which the production of vitamin B12 was not revealed in the phenotypic assays. However, the detection of a few genes does not support that the complete biosynthetic pathway is present. Moreover, their occurrence in the genome does not mean that they are being expressed. Finally, it is important to notice that except for *Fur. rossiae* DSM 15914, which was grown in microaerophilia, *Lpb. plantaram* G12 and the other tested isolates were grown in aerobiosis. Assuming that the presence of *cbiT* in the genome is evidence that cobalamin production follows an anaerobic pathway, bacteria growth conditions could explain the negative results obtained in the microbiological assay for *Lpb. plantarum* G12 and the low output observed for *Lac. lactis* E32 and *Lev. brevis* G31.

### 3.4. Lactic Acid Bacteria Probiotic Potential

The probiotic potential of *Lac. lactis* E32, *Lev. brevis* G31, *Lpb. plantarum* G12, and *P. pentosaceus* L51 was assessed considering their capacity to withstand the harsh gastrointestinal conditions, along with the absence of virulence and antibiotic resistance determinants.

#### 3.4.1. Evaluation of In Vitro Gastrointestinal Survival

*Lactococcus lactis* E32 and *Lev. brevis* G31 presented a similar resistance pattern upon exposure to harsh GIT conditions (Figure 2A,B). Despite the scarce impact of the gastric phase on the cell viability, an accentuated decline of 3.50 log CFU mL^−1^ (*Lac. lactis* E32) and 3.19 log CFU mL^−1^ (*Lev. brevis* G31) was obtained in the intestinal phase, indicating that bile had a significant impact on bacterial cells survival (*p* < 0.05).

*Lactiplantibacillus plantarum* G12 and *P. pentosaceus* L51 also demonstrated similar behaviors when submitted to the simulated digestion conditions (Figure 2C,D). Nonetheless, both bacterial strains have proved to be more resilient than *Lac. lactis* E32 and *Lev. brevis* G31. Viable counts of *Lpb. plantarum* G12 and *P. pentosaceus* L51 decreased 0.21 log CFU mL^−1^ and 0.58 log CFU mL^−1^, respectively, during the complete digestion process.

In the present work, the results indicated that *Lpb. plantarum* G12 was highly tolerant to gastric and intestinal stress factors, which is not in accordance with previous works [47,48,49]. However, one must consider that the former studies were performed at lower pH values (when compared with our experiments) and that the most accentuated reductions in cell viability essentially occurred at pH values lower than three.

Concerning the viability of *P. pentosaceus* L51 throughout the digestion assay, a total reduction of 0.6 log CFU mL^−1^ was obtained, which is in agreement with the previously documented by other authors [50,51]. Damodharan et al. [50] evaluated the GIT resistance of *P. pentosaceus* strain KID7, having observed a slight reduction of 1 log CFU ml^−1^ at the end of the whole digestion process. The survival of *P. pentosaceus* L1 was assessed by Cao et al. [51] and after 3 h of gastric juice (pH 2) exposure, a reduction of only <0.3 log CFU mL^−1^ was obtained. A similar value was observed after 4 h of exposure to the intestinal fluid.

In former studies, a pH value of two or lower was demonstrated to have a significant impact on the viability of *Lac. lactis*, while exposure to pH 3 originated a negligible cell reduction [52,53,54], which is in agreement with the results observed herein. Moreover, previous works provided evidence that the intestinal phase was a stress factor for *Lac. lactis* survival, which was corroborated by the results attained herein, with a reduction higher than 3 log CFU mL^−1^ being observed [53,54]. Similarly, a significant decrease (>3 log CFU mL^−1^) was observed in *Lev. brevis* G31, presenting the lowest tolerance to harsh gastrointestinal conditions at the end of digestion.

#### 3.4.2. Evaluation of the Safety Profile of the B12-Producing Strains

Despite strain-specific characteristics, lactic acid bacteria safety profile should also be considered when evaluating their suitability as probiotic candidates.

##### Antibiotic Susceptibility Evaluation

Concerning the antibiotic susceptibility profile of the isolated LAB, with the exception of vancomycin, to which *Lpb. plantarum* G12 and *P. pentosaceus* L51 were resistant, all the strains were sensitive to all the other antibiotics tested (Table 4). However, susceptibility to vancomycin is not required according to EFSA recommendations [35] since intrinsic resistance to this antibiotic is widespread in these species [55,56]. Resistance to tetracycline has been reported in many LAB species isolated from poultry [57]. In the current work, LAB were isolated from a microalgae culture-associated microbiota, which may explain the susceptibility to this antibiotic. Moreover, according to the complementary genomic analysis, the obtained antibiotic phenotypes were consistent with the bacterial genotypes. Therefore, the novel probiotic candidates have proven potential to be used for nutritional purposes without constituting a risk for antibiotic resistance spread to opportunistic pathogens [58].

##### Virulence Genes, Hemolytic Activity, and Biogenic Amine Production Evaluation

In the current work, the incidence of virulence factors involved in adhesion and aggregation (*esp*, *efaAfs*, *efaAfm*, and *agg*), biofilm formation (*gelE*), and production of hemolysin-cytolysin (*cylA*, *cylB*, *cylM*, *cylLL*, and *cylLS*) was investigated. All the virulence determinant genes were not detected for the four tested strains, corroborating the whole-genome prediction. The absence of these specific virulence determinants was in agreement with the formerly reported concerning other strains of *Lac. lactis* [59], *Lev. brevis* [60], *Lpb. plantarum* [61], and *P. pentosaceus* [62], and further supports the non-pathogenicity of the evaluated strains and, hence, the fulfillment of this safety prerequisite.

Moreover, nonhemolytic activity is also considered a pivotal criterion for the selection of a probiotic strain. *Lactococcus lactis* E32, *Lev. brevis* G31, and *P. pentosaceus* L51 were revealed to have a negative hemolytic phenotype, being considered γ-hemolytic, while *Lpb. plantarum* G12 showed to be α-hemolytic, presenting partial hemolytic capacity (Table 5). In accordance with this latter result, an α-hemolysis phenotype has been formerly documented for other *Lpb. plantarum* strains [63,64,65,66].

The production of biogenic amines by LAB, including tyramine, histamine, cadaverine, and putrescine, is also an important aspect to consider when evaluating their probiotic potential. Although LAB have been reported as the main biogenic amine producers in fermented foods [67], in the current work none of the selected strains produced the biogenic amines investigated (Table 5). This is a desirable trait since the presence and levels of these biogenic amines in fermented foods are of significant concern from a food safety perspective [68].

Taken as a whole, the in vitro safety profile of *Lac. lactis* E32, *Lev. brevis* G31, and *P. pentosaceus* L51 may support the GRAS status of these novel strains, indicating their suitability for human consumption.

## 4. Conclusions

In the present work, three novel bacterial strains, *Lac. lactis* E32, *Lev. brevis* G31, and *P. pentosaceus L51*, were identified as potential cobalamin producers. To the authors’ knowledge, this is the first work documenting a *Lac. lactis* strain capable of producing vitamin B12. Noteworthy, considering the highest capacity for de novo cobalamin biosynthesis and remarkable probiotic features, *P. pentosaceus* L51 was found to be the most promising candidate strain for developing high-content vitamin B12 formulations. Nonetheless, in order to fully evaluate the safety profile of this B12-producing strain and its potential health-promoting benefits, additional in vivo experiments ought to be performed.

## Figures and Tables

**Figure 1 foods-12-03277-f001:**
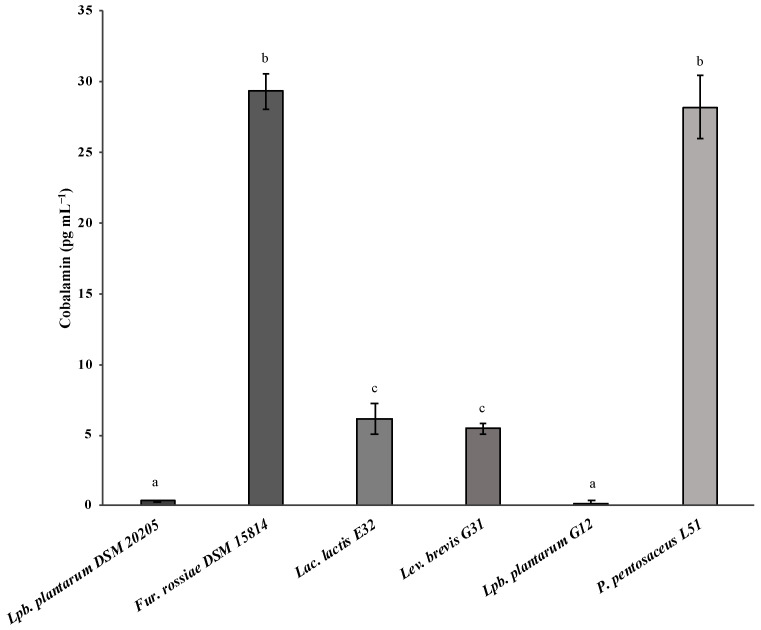
Cobalamin production by *Lac. lactis* E32, *Lev. brevis* G31, *Lpb. plantarum* G12, and *P. pentosaceus* L51. *Lpb. plantarum* DSM 20205 and *Fur. rossiae* DSM 15814 were used as a negative and positive control, respectively. Values represent the mean of three independent experiments and the error bars represent standard deviations of the mean values. Different letters indicate significant differences (*p* < 0.05).

**Figure 2 foods-12-03277-f002:**
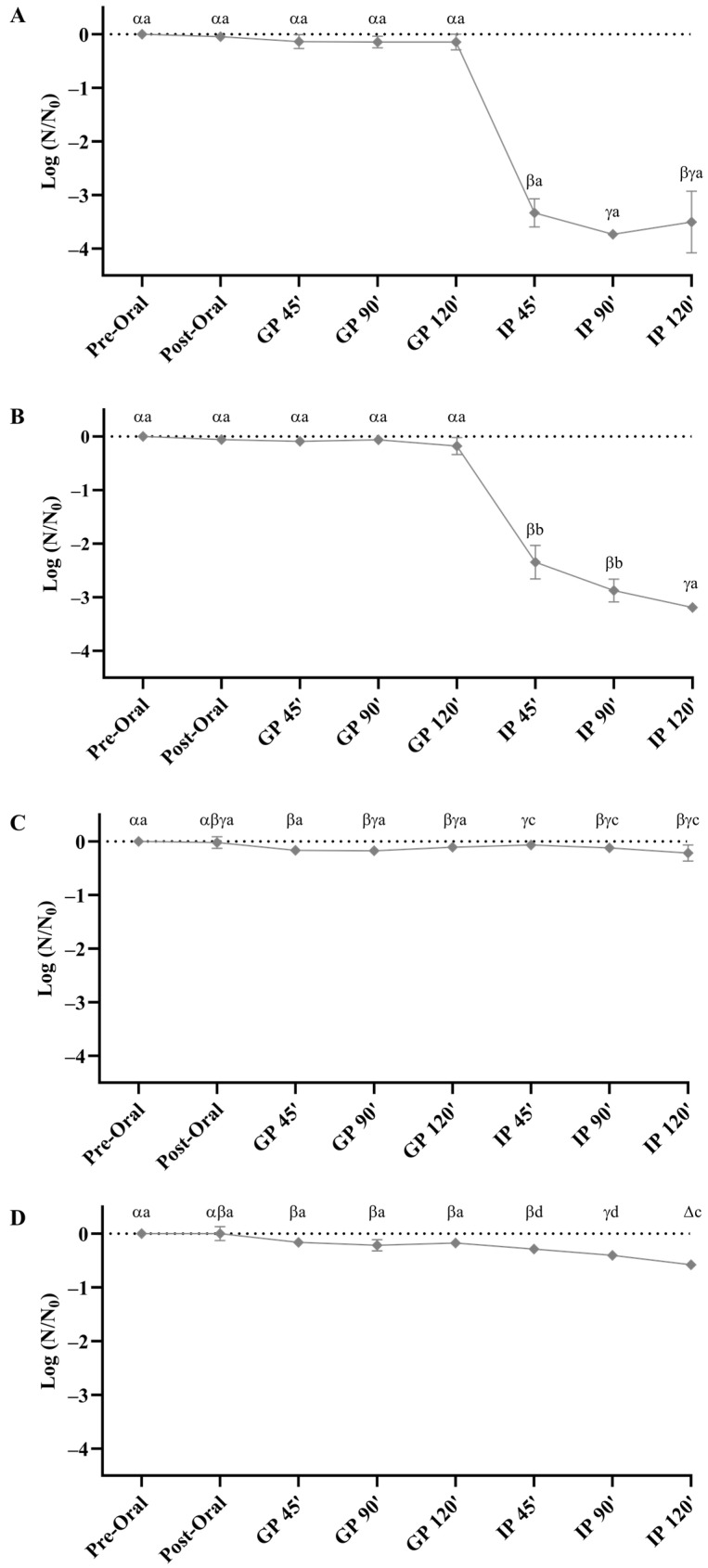
Evaluation of the effect of simulated gastrointestinal digestion on the viability of *Lac. lactis* E32 (**A**), *Lev. brevis* G31 (**B**), *Lpb. plantarum* G12 (**C**), and *P. pentosaceus* L51 (**D**). N_0_ is the initial bacterial density in CFU per milliliter, and N is bacterial density at a particular phase of gastrointestinal digestion (GP, gastric phase; IP, intestinal phase). Values represent the mean of three independent experiments and the error bars represent standard deviations of the mean values. Different lowercase letters indicate statistical significance (*p* < 0.05) (Greek and Latin lowercase letters refer to differences between logarithmic reductions at the predefined time points, within the same bacterial population and between the four different bacterial populations, respectively).

**Table 1 foods-12-03277-t001:** Identification of lactic acid bacteria strains isolated from *C. vulgaris* bioreactors.

Bacterial Strains	Number of Isolates
*Lactococcus lactis*	95
*Enterococcus casseliflavus*	16
*Lactiplantibacillus plantarum*	14
*Pediococcus pentosaceus*	3
*Levilactobacillus brevis*	2

**Table 2 foods-12-03277-t002:** Examples of vitamin B12-producing lactic acid bacteria and the corresponding content of the biosynthesized vitamin.

Lactic Acid Bacteria	Vitamin B12 Content	Reference
*Lev. brevis*	0.6 pg mL^−1^	[21]
*P. pentosaceus*	0.4–0.6 pg mL^−1^	[21]
*Lpb. plantarum* CN-225	2.0 pg mL^−1^	[21]
*Lpb. plantarum*	13 ng of g^−1^ DW	[42]
*Lpb. plantarum* BHM10	0.5–0.8 pg mL^−1^	[24]
*Lpb. plantarum* BCF20	0.5–0.8 pg mL^−1^	[24]
*Lac. lactis* E31	6.18 ± 1.08 pg mL^−1^	This work
*Lev. brevis* G31	5.47 ± 0.37 pg mL^−1^	
*P. pentosaceus* L51	28.19 ± 2.27 pg mL^−1^	

**Table 3 foods-12-03277-t003:** Presence of the cobalamin biosynthesis genes in *Lac. lactis* E32, *Lev. brevis* G31, *Lpb. plantarum* G12, and *P. pentosaceus* L51 genomes. The complete cobalamin cluster comprises *gltX*, *hemA*, *hemL*, *hemB*, *hemC*, and *hemD* genes for the uroporphyrinogen III synthesis; *cobA*, *cysG*, *cbiK*, *cbiL*, *cbiH*, *cbiF*, *cbiG*, *cbiD*, *cbiJ*, *cbiE*, *cbiT*, *cbiC*, *cbiA*, *cobR*, *cobO*, *cobQ*, *cobD*, and *cbiB* for adenosylcobinamide synthesis; and *cobU*, *cobT*, *cobS*, and *cobC* for lower ligand synthesis.

	*Lac. lactis* E32		*Lev. brevis* G31		*Lpb. plantarum* G12		*P. pentosaceus* L51	
	E-Value	Identity	E-Value	Identity	E-Value	Identity	E-Value	Identity
* **hemL** *	4.00 × 10^−24^	27%	4.00 × 10^−24^	27%	1.00 × 10^−28^	31%	0	0%
* **cobA** *	0	0%	0	0%	0	0%	4.00 × 10^−28^	38%
* **cbiT** *	7.00 × 10^−5^	26%	2.00 × 10^−5^	26%	2.00 × 10^−4^	26%	7.00 × 10^−4^	27%
* **cobD** *	4.00 × 10^−9^	23%	3.00 × 10^−9^	23%	8.00 × 10^−17^	26%	1.00 × 10^−8^	32%
* **cobC** *	9.00 × 10^−7^	30%	9.00 × 10^−7^	30%	6.00 × 10^−9^	30%	3.00 × 10^−6^	24%

**Table 4 foods-12-03277-t004:** Antibiotic susceptibility of *Lac. lactis* E32, *Lev. brevis* G31, *Lpb. plantarum* G12, and *P. pentosaceus* L51 strains; MIC and cut-off values are presented in mg L^−1^.

LAB Strains		Antibiotic Susceptibility								
		AMP	CHL	CLI	ERY	GEN	KAN	STR	TET	VAN
*Lac. lactis* E32	*MIC*	0.25	4	0.5	0.5	32	32	32	0.5	0.5
	*Cut-off*	2	8	8	1	32	64	32	4	4
*Lev. brevis* G31	*MIC*	0.25	4	0.5	0.5	16	32	64	0.25	0.5
	*Cut-off*	2	4	4	1	16	32	64	8	n.r.
*Lpb. plantarum* G12	*MIC*	0.25	4	8	0.5	8	64	32	8	R.
	*Cut-off*	2	8	8	1	16	64	n.r.	32	n.r.
*P. pentosaceus* L51	*MIC*	2	4	0.125	0.25	16	64	64	8	R.
	*Cut-off*	4	4	4	1	16	64	64	8	n.r.

AMP—ampicillin; CHL—chloramphenicol; CLI—clindamycin; ERY—erythromycin; GEN—gentamicin; KAN—kanamycin; STR—streptomycin; TET—tetracycline; VAN—Vancomycin; n.r.—not required; R—resistant.

**Table 5 foods-12-03277-t005:** Hemolytic activity and biogenic amine production evaluation.

LAB Strain	Haemolysis	Biogenic Amine-Producer Phenotype			
		Tyramine	Histamine	Cadaverine	Putrescine
*Lac. lactis* E32	γ	N	N	N	N
*Lev. brevis* G31	γ	N	N	N	N
*Lpb. plantarum* G12	α	N	N	N	N
*P. pentosaceus* L51	γ	N	N	N	N

N—negative biogenic amine-producer phenotype.

## Data Availability

The data presented in this study are available on request from the corresponding author.

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
