# Peer review of "Exploiting Potential Probiotic Lactic Acid Bacteria Isolated from Chlorella vulgaris Photobioreactors as Promising Vitamin B12 Producers"

_foods, 2023, doi:10.3390/foods12173277_

Round 1

Reviewer 1 Report (Previous Reviewer 1)

The manuscript has been significantly improved, the authors have clearly specified the aim of the work and the conclusions are know well-validated by the results presented. Thus, I consider the manuscript is suitable for publication after some minor corrections which are indicated in the attached manuscript pdf (highlighted in lightblue).

Author Response

Response: We are grateful to the reviewer for the positive feedback and for drawing attention to the two indicated incorrections. The text was revised accordingly.

Reviewer 2 Report (Previous Reviewer 2)

I would like to thank the authors for having responded to my comments.

Author Response

Response: We are grateful to the reviewer for the positive feedback.

Reviewer 3 Report (Previous Reviewer 3)

Dear Editors and authors, 

Major comments

1-The work methods in the manuscript are vague and unclear, some of them do not contain scientific references that can be consulted when reading them, and there are work methods without results.

Minor comments

1-A method of isolating lactic acid bacteria and growing it on different media for which no scientific reference was mentioned. See line 89-98.

2- Where are the results of these methods? ((The obtained isolates were examined by Gram reaction and catalase activity test. The cell morphology was inspected under an optical microscope)).

3-The main work method in the manuscript does not include a scientific reference that can be read and consulted.

The manuscript is based on the production of vitamin B12, but the method of production is without scientific reference.

4-How do I take vitamin B12? The method used is a colorimetric method that cannot be relied upon. The researchers should have used other methods of estimation, such as HPLC .

5-In vitro gastrointestinal survival analysis, The authors explain the method of studying the conditions of the digestive system, this method is vague and unclear, how the author used Buffer solution with a pH of 7.4 and that the pH of the stomach is 1.5-5????

6-The statistical analysis did not appear in any of the results in the tables or figures, except for Figure 1.

7-What is the scientific basis for selecting 130 isolates?

8-How many bacterial isolates did the authors have? The authors indicated that 20% of the isolates appearing in the dishes were selected.

 9-The results of the amount of vitamin produced are incorrect because they use the spectrophotometer method for estimation. In this method, two peaks appear and there is an overlap between the two peaks, which affects the results, and the authors did not mention the method clearly in the work methods.

10- Enterococcus faecalis Bacteria were mentioned in the work methods because they were tested for vitamin production and antibiotic resistance, but these bacteria did not show any results in the tables and figures. Why????

11-There is no figure or table showing the types of  biogenic amines  that were tested and the results of haemolytic activity of bacteria.

12-The conclusions are very weak, and it is results, not conclusions

The language of the manuscript is good.

Author Response

Major comments

1-The work methods in the manuscript are vague and unclear, some of them do not contain scientific references that can be consulted when reading them, and there are work methods without results.

Response: The authors acknowledge the reviewer for the comment. Hence, following the reviewer’s suggestion, we have now included additional references to describe the methodologies utilized in the current study.

Minor comments

Point 1: A method of isolating lactic acid bacteria and growing it on different media for which no scientific reference was mentioned. See line 89-98.

Response to Point 1: The authors thank the reviewer for the comment. In this sense, following the reviewer’s suggestion, we have now included a reference (Line 95).

Point 2: Where are the results of these methods? (The obtained isolates were examined by Gram reaction and catalase activity test. The cell morphology was inspected under an optical microscope).

Response to Point 2: The authors thank the reviewer for the comment. As we previously mentioned, LAB are characterized as Gram-positive and catalase-negative bacteria. Nonetheless, following the reviewers suggestion, we have now included this information (Line 214). Moreover, the cell morphology description was also included (Lines 216-218).

Point 3: The main work method in the manuscript does not include a scientific reference that can be read and consulted. The manuscript is based on the production of vitamin B12, but the method of production is without scientific reference.

Response to Point 3: The authors thank the reviewer for drawing attention to the relevance of the inclusion of a reference. Hence, following the reviewer’s suggestion, we have now included additional references to describe the method (Lines 115 and 128).

Point 4: How do I take vitamin B12? The method used is a colorimetric method that cannot be relied upon. The researchers should have used other methods of estimation, such as HPLC.

Response to Point 4: While we appreciate the reviewer’s comment, we must clarify that in the current work the vitamin B12 content was evaluated by a microbiological assay. For this purpose, Lactobacillus leichmannii DSM 20355 (formerly Lactobacillus delbrueckii subsp. lactis) was utilized as an indicator strain (requiring vitamin B12 for growth) to evaluate the cobalamin content (hence, what was determined spectrophotometrically (OD600) was the bacterial growth).

Point 5: In vitro gastrointestinal survival analysis, The authors explain the method of studying the conditions of the digestive system, this method is vague and unclear, how the author used Buffer solution with a pH of 7.4 and that the pH of the stomach is 1.5-5????

Response to Point 5: We thank the reviewer for the comment. However, we must clarify that in the current work, the in vitro gastrointestinal survival analysis was performed according to the INFOGEST method (static in vitro simulation of gastrointestinal food digestion) described by Brodkorb (2019). Hence, to mimic the gastric phase, a simulated gastric fluid of pH 3.0 was prepared according to the mentioned reference. As described in the manuscript, phosphate buffer saline (PBS, pH 7.4) was only utilized to dilute the aliquots collected at predefined time intervals in order to perform the enumeration of viable bacterial cells.

Point 6: The statistical analysis did not appear in any of the results in the tables or figures, except for Figure 1.

Response to Point 6: We thank the reviewer’s comment.  In order to address the reviewer´s concerns, the statistical analysis of the in vitro gastrointestinal survival experiments (the other experiment in which statistical analysis was performed) was included and, hence, Figure 2 was revised accordingly.

Point 7: What is the scientific basis for selecting 130 isolates?

Response to Point 7: We thank the reviewer for the question. The 130 isolates were selected on the basis of representing 20% of total colonies from countable agar plates, which is what is conventionally utilized in this microbiological methodology. A bibliographical reference was added accordingly (Line 96).

Point 8: How many bacterial isolates did the authors have? The authors indicated that 20% of the isolates appearing in the dishes were selected.

Response to Point 8: We thank the reviewer for the question. The total number of bacterial isolates was 650.

Point 9: The results of the amount of vitamin produced are incorrect because they use the spectrophotometer method for estimation. In this method, two peaks appear and there is an overlap between the two peaks, which affects the results, and the authors did not mention the method clearly in the work methods.

Response to Point 9: We thank the reviewer for the question. Nonetheless, we must clarify that (as abovementioned, in “Response to Point 4”) in the current work the selected LAB isolates were screened for their vitamin B12 biosynthesis capacity following a microbiological approach for cobalamin quantification.

Point 10: Enterococcus faecalis Bacteria were mentioned in the work methods because they were tested for vitamin production and antibiotic resistance, but these bacteria did not show any results in the tables and figures. Why????

Response to Point 10: We thank the reviewer for the question. Nonetheless, we must clarify that Enterococcus faecalis ATCC 29212 was only used as a quality control reference strain for the antibiotic susceptibility evaluation according to the Clinical and Laboratory Standards Institute (CLSI) and European Committee on Antimicrobial Susceptibility Testing (EUCAST) guidelines. The sentence was revised accordingly (Line 179).

Point 11: There is no figure or table showing the types of  biogenic amines that were tested and the results of haemolytic activity of bacteria.

Response to Point 11: The authors thank the reviewer for the suggestion. Hence, we have now included Table 5 to present those results.

Point 12: The conclusions are very weak, and it is results, not conclusions.

Response to Point 12: The authors thank the reviewer for the comment. However, we respectfully disagree with the comment, since the mentioned section does present conclusions drawn from the results obtained. Nonetheless, in order to address the reviewer´s concerns, an additional sentence was added at the end of the section.

Round 2

Reviewer 3 Report (Previous Reviewer 3)

Dear Editors and authors, 

Many of the answers are not convincing, for example, how to select 130 bacterial isolates and what is the scientific basis for that?

The answer to the question was not mentioned in the manuscript.

The number of total isolates was, according to the authors' answer, 650 isolates, and this was not mentioned in the manuscript, and many other answers.

The manuscript cannot be accepted with these many contradictions.

The language of the manuscript is good and clear to the reader. 

Author Response

Thanks  for these final comments.   *The 130 isolates were selected on the basis of representing 20% of total colonies from countable agar plates, which is what is conventionally utilized in this microbiological methodology. A bibliographical reference was added accordingly   *650 was included in the manuscript.

This manuscript is a resubmission of an earlier submission. The following is a list of the peer review reports and author responses from that submission.

Round 1

Reviewer 1 Report

In the present work, the authors study four LAB strains isolated from C. vulgaris culture photobioreactor and characterized them on account of their safety and ability to produce vitamin B12. In the introduction, the authors emphasized the relevance of increasing the vitamin B12 content in the microalga C. vulgaris through co-culture with Vit B12-producing LAB and they concluded that P. pentosaceus L51 is “a promising candidate strain for developing high-content vitamin B12 formulations, namely in C. vulgaris biomass”. It is not clear which is the B12 bioenriched product that the authors propose? Is it formulated with a co-culture of P. pentosaceus L51- C. vulgaris? Does the presence of P. pentosaceus L51 improve the growth of the microalga or the production of Vit B12? The authors did not study the effect of  P. pentosaceus L51 on C. vulgaris, so the conclusion is not validated by the results presented in the work. For this reason, I consider that the manuscript requires specific studies for the potential application of Vitamin B12 producing LAB strains in C. vulgaris biomass in order to validate the hypothesis propose by the authors.

Moreover, safety results (antibiotic susceptibility, virulence factors, biogenics amines) are only mentioned in the text. I consider some of these results could be included in tables or figures in order to improve the manuscript. For example, a table indicating the MIC for each strain against different antibiotic (with the corresponding cut off) could be added. Discussion must be improved, particularly on account of the potential application of these strains in the context of C. vulgaris culture.

Reviewer 2 Report

The paper entitled "Exploiting Potential Probiotic Probiotic Lactic Acid Bacteria Isolated from 2 Chlorella vulgaris Photobioreactors as Promising Vitamin B12 3 Producers", aims to "evaluate the vitamin B12 biosynthesis capacity of LAB isolated from C. vulgaris bioreactors and characterize their probiotic potential". It seems to me a well-written and scientifically rigorous document. I consider that the authors did an exhaustive work in the evaluation of the strains and their results are pertinent.

I have only one observation to improve the document:

The caption of figure 1 mentions "Different letters indicate significant 224 differences (p<0.05)", however, the figure does not show any letter.

Reviewer 3 Report

Dear Editors and authors,

The manuscript (Exploiting Potential Probiotic Lactic Acid Bacteria Isolated from Chlorella vulgaris Photobioreactors as Promising Vitamin B12 Producers) has good idea but needs some corrects. 

1- The aim of the manuscript is unclear, please rewrite again.

2-Some method within references, See line 99-102,  I suggest you  to add 

 Niamah, A. K., Abdulmuttaleb, A. M., & Hussain, N. A. (2020). ANTIBACTERIAL SPECTRUM OF PRODUCED REUTERIN FROM NEW ISOLATES OF Lactobacillus reuteri. Journal of microbiology, biotechnology and food sciences, 10(1), 134-139.

See 112-132, 

Walhe, R. A., Diwanay, S. S., Patole, M. S., Sayyed, R. Z., Al-Shwaiman, H. A., Alkhulaifi, M. M., ... & Datta, R. (2021). Cholesterol reduction and vitamin B12 production study on Enterococcus faecium and Lactobacillus pentosus isolated from yoghurt. Sustainability, 13(11), 5853.

3- polymerase chain 107 reaction (PCR) assay for the amplification of 16S rRNA gene (Line 107-108) Please add the Program of PCR.

4-Some tests are mentioned in the work methods, such as Gram staining, cell morphology, and the catalase enzyme test (See line 101-102) , but there are no results for them in the results chapter. Why?

5-Figure 1 The X-axis label should be added. I suggest it be (lactic acid bacteria isolates)

6-There is ambiguity in the results of Table 1 and Figure 1. The authors mention the number of isolates as 130 isolates. Four isolates were chosen on the basis of vitamin production. Where are the results of the rest of the isolates and why were they not mentioned in the table or in the discussion?

7-Table 3 Some results were left blank, and this is not a result. A number must be mentioned, even if it is 0. The table must be modified.

8- Figure 2, what do you value in N and N0, that was not mentioned in the work methods.  

9- Figure 2, there is no label for the x-axis in the figure!!!

10-The results of Antibiotic susceptibility evaluation are not clear and do not indicate which concentration is effective in the bacteria. The methods of action speak of Minimal inhibitory concentrations (see line 164). These results are vague and unclear.

11-The results of Virulence genes, haemolytic activity and biogenic amine production evaluation are not clear and do not show the genes present in each bacterium. This part of the work should be deleted.

12-Conclusions contain a lot of results. Results should be deleted and conclusions rewritten clearly.